# Identifying and reporting position-specific countermovement jump outcome and phase characteristics within rugby league

**John J. McMahon**[1]\*, **Jason P. Lake**[2☯], **Paul Comfort**[1,3,4☯]

**1** Directorate of Psychology and Sport, University of Salford, Salford, United Kingdom, **2** Chichester Institute of Sport, University of Chichester, Chichester, West Sussex, United Kingdom, **3** Institute for Sport, Physical Activity and Leisure, Carnegie School of Sport, Leeds Beckett University, Leeds, United Kingdom, **4** Centre for Exercise and Sport Science Research, Edith Cowan University, Joondalup, Australia

☯ These authors contributed equally to this work.
\* j.j.mcmahon@salford.ac.uk

**Data Availability Statement:** The raw data from the countermovement jump test is available as a Supporting information document. The document does not contain any personal identifying

## Abstract

The countermovement jump (CMJ) has been suggested to be an important test of neuromuscular performance for rugby league (RL) players. Identifying force platform-derived CMJ variables that may be more applicable to RL positions (e.g., forwards and backs) has yet to be fully explored in the scientific literature. The aim of this study was to identify RL position-specific CMJ force-time variables. Specifically, we aimed to compare select force-time variables from the countermovement (i.e., the combination of unweighting and braking) and propulsion phases of the CMJ between the global forwards and backs positional groups. We also aimed to compare typical (i.e., jump height) and alternative (i.e., take-off momentum) outcome CMJ variables between positional groups. Finally, we sought to visually present each individual player's CMJ performance alongside the average data to facilitate the interpretation and reporting of the CMJ performances of RL athletes. Twenty-seven forwards and twenty-seven backs who competed in the senior men's English RL Championship, performed three CMJs on a force platform at the beginning of the pre-season training period. There were no significant differences in any countermovement or propulsion phase variable between positions with just small effect sizes noted ($P \geq 0.09$, $d \leq 0.46$). Jump height (and so take-off velocity) was significantly greater for backs with moderate effects displayed ($P = 0.03$, $d = 0.60$). Take-off momentum (take-off velocity × body mass) was largely and significantly greater for forwards ($P < 0.01$, $d = 1.01$). There was considerable overlap of individual player's body mass and CMJ outcome variables across positions, despite significant differences in the mean values attained by each positional group. The results suggest that it may be beneficial for RL practitioners to identify player-specific, or at least position-specific, variables. As a minimum, it may be worthwhile selecting CMJ force-time variables based on what is considered important to individual player's or small clusters of similar players' projected successes during RL competition.

information about the participants, just the raw
data used for the study which is split into the two
positional groups tested.

**Funding:** The author(s) received no specific
funding for this work.

**Competing interests:** The authors have declared
that no competing interests exist.

## Introduction

The countermovement jump (CMJ) has been suggested to be an important test of neuromuscular performance for rugby league players [1]. This suggestion is supported by researchers who reported that greater CMJ heights were correlated with faster 5-, 10- and 30 m sprint performances [2] and better tackling ability [3] in high-level players ($r = 0.38$–$0.62$). Additionally, players selected in the first Australian National Rugby League game of the season demonstrated, among other factors, superior jump heights compared to non-selected players [4]. Furthermore, it was reported that greater CMJ height and reactive strength index modified ($RSI_{mod}$, calculated as CMJ height divided by time to take-off), which were derived from a force platform using criterion methods, discriminated between senior and academy (under 19s, the final academy age group before senior-level) rugby league players competing in the English Rugby League Championship (RLC, second tier of English Rugby League) [5]. The time to take-off was similar between the groups but senior players jumped higher (Cohen's $d$ [$d$] = 0.91) which led to them achieving a higher $RSI_{mod}$ ($d = 0.58$) [5]. Conversely, senior players who competed in the English Super League (SL, top tier of English Rugby League) achieved a slightly greater CMJ height ($d = 0.12$ [trivial effect]), but a moderately greater mean propulsion power ($d = 0.94$) than the u19s players' scores [6]. This was likely due to the senior players performing the CMJ with a shorter propulsion phase time, although this data were not reported [6]. Similarly, senior SL players demonstrated a greater $RSI_{mod}$ ($d = 1.11$) than senior RLC players, owing to their capacity to attain a similar jump height ($d = 0.25$) but via a much shorter time to take-off ($d = 1.26$) value [7], highlighting a superior neuromuscular performance.

With respect to the two main positional groups within rugby league, backs jumped higher than forwards by ~2 cm (Hedge's $g$ [$g$] = 0.44) and attained a higher $RSI_{mod}$ by ~0.03 units ($g = 0.37$) in a recent study in which researchers assessed combined senior SL and senior RLC rugby league players' ($n = 104$) CMJ performances via a force platform [8]. Both mean and peak relative propulsion power (presented as Watts per kilogram of body mass) were moderately greater ($g = 0.65$–$0.70$) for the backs too [8]. If one were to take these results at face value, it would be concluded that backs outperform forwards in the CMJ. Of course, this statement could only be made with reference to these variables, but these are the most reported CMJ variables in rugby league studies [5, 6, 9]. However, the forwards were, on average, ~12 kg heavier than the backs ($g = 1.34$) which represented a large effect. Nevertheless, forwards only produced a slightly poorer jump height, $RSI_{mod}$ and relative power (small-moderate effect size) in the CMJ. As body mass impedes take-off velocity, and therefore jump height, did the forwards really perform inferiorly to the backs? As described in the previous study [8], jump height, $RSI_{mod}$, mean propulsion power and peak propulsion power are all biased towards lighter athletes (i.e., generally the backs). Thus, identifying position-specific force platform-derived CMJ variables that may be more applicable to collision sport athletes, who are sometimes required to be able to effectively accelerate their own body mass and the body masses of their opponents, is warranted.

In a collision sport such as rugby league, sprint momentum (which includes body mass in its calculation: body mass × velocity) has been highlighted as a more important factor than sprint velocity and associated variables [10]. This is, perhaps, why sprint momentum is now more commonly reported by rugby league researchers and practitioners than sprint velocity or split times. A rationale for calculating jump take-off momentum as part of routine CMJ testing of rugby league players was recently presented [11]. Researchers showed that jump take-off momentum and sprint momentum attained at 5, 10 and 20 m demonstrated very large-near perfect associations ($r = 0.78$–$92$; 61–85% shared variance [$R^2$]) in a high-level (RLC) rugby league cohort [11]. This preliminary study included CMJ and sprint data from of a mixture of

forwards (n = 14) and backs (n = 11), so there is currently no position-specific data available on jump take-off momentum within rugby league or, indeed, any other collision sports. A more recent study illustrated that the high correlation between sprint and jump take-off momentum is a product of body mass being included in each calculation and is influenced particularly when there is a large variance in body mass scores and a low variance in jump height scores [12], as is seen in rugby league cohorts. As already mentioned, being heavier can be considered an asset for certain positions within rugby league but this should not detract from the importance of being able to attain high velocities too, either to achieve an even larger momentum, to beat an opponent to the ball or to break away from the opposition. Indeed, this may be why, as mentioned earlier, greater CMJ heights were correlated with better tackling ability [3].

Currently, there is limited information available that can be gleaned from empirical studies to aid rugby league researchers' and practitioners' identification of practically meaningful and position-specific force platform derived CMJ metrics. Given that specific CMJ variables can discriminate performance level in rugby league [4–7] and it is recommended that CMJ testing of this cohort should ideally be performed using a force platform [13, 14], there is a requirement to fill this knowledge gap. Additionally, although jump take-off momentum may be a promising metric for collision-sports, only one study has reported it to date and the data were obtained from an entire rugby league cohort rather than from positional groups [11]. Also, there are currently no criteria to establish whether a rugby league player's body mass effectively contributes to their jump take-off momentum. For example, rugby league players who are heavy but achieve lower than expected jump take-off momentum may benefit from either reducing their body mass slightly to be able to move faster or developing sufficient strength to be able to move their body mass faster. Plotting the relationship between body mass and jump take-off momentum and incorporating a quadrant chart to illustrate the cause-effect relationship (i.e., with the quadrant comprised of 1) low body mass with low momentum, 2) low body mass with high momentum, 3) high body mass with low momentum, and 4) high body mass with high momentum) should reveal which rugby league player's body mass is contributing effectively to their jump take-off momentum.

The aim of this study was to identify position-specific CMJ force-time variables within rugby league. Specifically, we aimed to compare select force-time variables from the countermovement (i.e., the combination of unweighting and braking) and propulsion phases of the CMJ between the global forwards and backs positional groups. We also aimed to compare typical (i.e., jump height) and alternative (i.e., take-off momentum) outcome CMJ variables between positional groups. It was hypothesised that absolute kinetic variables would be larger for the forwards and relative (to body mass) kinetic variables would be larger for the backs, owing to the larger body mass of the forwards. Consequently, it was hypothesised that jump height would be higher for the backs, but take-off momentum would be larger for the forwards. Finally, despite the previously mentioned hypotheses, that were formulated due to expected differences in the average body mass of each positional group, it was further hypothesised that there would be considerable overlap of individual player's body mass and CMJ performances across the forwards and backs. Therefore, we also sought to visually present each individual player's CMJ performance alongside the average data to help inform how rugby league practitioners may best interpret and report the CMJ performances of their athletes.

## Materials and methods

A convenience sampling approach led to twenty-seven forwards (age = 25.9 ± 4.2 years, height = 1.85 ± 0.06 m, body mass = 102.9 ± 9.4 kg) and twenty-seven backs (age = 25.9 ± 3.5 years, height = 1.81 ± 0.06 m, body mass = 88.7 ±7.9 kg), who at the time of testing competed

in the senior men's English RLC, participating in this study. All players were free from injury and engaged in a full-time strength and conditioning programme (strength and power focussed, after a period of hypertrophy focussed training) at the time of testing (the start of the pre-season). Written informed consent was provided prior to testing, the study was pre-approved by the University of Salford institutional review board and conformed to the World Medical Association's Declaration of Helsinki.

Following a brief (~10 minutes) warm-up comprised of dynamic stretching and sub-maximal jumping (5×1 sets of single effort and 2×5 repeated CMJs), participants performed three recorded maximal effort CMJs to their preferred countermovement depth, each interspersed by ~1 minute rest [15]. They were instructed to "jump as fast and as high as possible", whilst always keeping their hands on their hips. The participants were informed that the "jump as fast" part of the verbal cue referred to them performing the countermovement and propulsion phases of the jump as quickly as possible. Verbal cues were standardised owing to their influence of CMJ force-time characteristics [16, 17].

Ground reaction forces during the maximal effort CMJs were sampled at 1000 Hz using a Kistler type 9286AA force platform and Bioware 5.11 software (Kistler Instruments Inc., Amherst, NY, USA). The force platform was zeroed before each CMJ trial. The participants stood upright and still for the first second of data collection [18, 19] to enable calculation of body weight (N, vertical force averaged over 1 s) and body mass (kg, body weight divided by gravitational acceleration). Raw vertical force-time data were exported as text files and analysed using a customized Microsoft Excel spreadsheet (version 2016, Microsoft Corp., Redmond, WA, USA). Previous research supports the analyses of raw CMJ force-time data [20].

Firstly, net force was calculated by subtracting body weight from every force sample. Centre of mass velocity was then determined by dividing net force by body mass on a sample-by-sample basis and then integrating the product using the trapezoid rule [19]. Instantaneous centre of mass displacement was determined by integrating the velocity data at each time point, again using the trapezoid rule [19]. The onset of movement was identified in line with current recommendations for unloaded CMJs [18]. The instant of take-off was identified when force fell below a threshold equal to five times the standard deviation (SD) of the flight phase force [21–23]. The SD of the flight phase force was calculated across the middle 50% of the flight phase duration when the force platform was unloaded [21, 22]. The CMJ phases were identified using the terminology explained recently by McMahon et al. [24]. Specifically, the countermovement phase comprised of the unweighting and braking phases and was defined as occurring between the onset of movement and zero velocity and the propulsion phase was deemed to have started when velocity exceeded $0.01 \text{ m·s}^{-1}$ (typically one sample after zero velocity) and finished at take-off [5, 25].

A selection of CMJ 'strategy' variables were calculated to help explain how the outcome variables were attained. Countermovement and propulsion phase times were calculated as the duration of each phase. Countermovement and propulsion displacement were calculated as the change in vertical centre of mass position between the start and end of each phase. Furthermore, countermovement and propulsion displacement (i.e., the displacement between the end of the braking phase and the instant of take-off) were also expressed as a percentage of standing centre of mass height which, in turn, was calculated as 57% of standing height [26]. This enabled a relative (to stature) between-group comparison [7]. Peak negative velocity, which occurs at the transition from unweighting to braking, was calculated to provide an indication of how fast the initial countermovement phase was performed. Net force at zero velocity was calculated as the net force at the end of braking phase, as this dictates the net force at the beginning of propulsion. Finally, propulsion mean net force was calculated as the average net force during the propulsion phase as this describes the force component that contributes to the propulsion net impulse.

Three CMJ outcome variables were calculated. Take-off velocity was calculated as the centre of mass velocity at the instant of take-off. Jump height was derived from vertical velocity at take-off [19]. The $RSI_{mod}$ was calculated as CMJ height divided by time to take-off (time between onset of movement to take-off). Jump take-off momentum was calculated by multiplying take-off velocity by the participant's body mass [11].

## Statistical analyses

A series of two-way mixed-effects model (average measures) intraclass correlation coefficients (ICC [type 3,$k$]), along with the upper and lower 95% confidence interval ($CI_{95}$), were used to determine the relative between-trial reliability of each variable. Based on the lower bound $CI_{95}$ of the ICC estimate, < 0.5, between 0.5 and 0.75, between 0.75 and 0.90 and >0.90 were indicative of poor, moderate, good and excellent relative reliability, respectively [27]. Absolute between-trial reliability of each variable was calculated using the coefficient of variation percentage (CV%), along with the upper and lower $CI_{95}$. A CV of ≤10% and ≤5% (based on the $CI_{95}$ of the CV% estimate) was considered to represent good and excellent absolute reliability, respectively [28].

All variables met parametric assumptions, apart from net force at zero velocity (both absolute and relative) and relative propulsion mean net force, following the Shapiro-Wilk test of normality. The mean differences between the forwards' and backs' normally distributed CMJ data were established using independent t-tests whereas non-parametric variables were compared between groups via the Mann-Whitney U test. Effect sizes were calculated (Cohen's $d$) and were interpreted as trivial (≤0.19), small (0.20–0.49), moderate (0.50–0.79), or large (≥0.80) [29]. A quadrant scatter chart was produced to illustrate the cause-effect relationship between take-off momentum and body mass, with a Pearson correlation test ($r$) performed to explore the magnitude of the relationship, along with the upper and lower $CI_{95}$. The correlation coefficient was interpreted as trivial (0.0–0.09), small (0.1–0.29, moderate (0.3–0.49), large (0.5–0.69), very large (0.7–0.89), and nearly perfect (0.9–1.0) [30]. The coefficient of determination ($R^2$) was calculated by squaring the correlation coefficients. Visual representations of the individual forward's and back's body mass, jump height and take-off momentum values, along with the mean values and mean differences, were also produced using Gardner–Altman plots [31]. Independent t-tests, the Mann-Whitney U tests, the Pearson correlation test, and ICCs were performed using SPSS software (version 25; SPSS Inc., Chicago, IL, USA) with the *a priori* alpha level set at $P \leq 0.05$. The effect sizes, CV% calculations and the quadrant scatter chart were produced in Microsoft Excel.

## Results

There was no difference in age between the forwards and backs ($P = 0.972$, $d = 0.01$), but the former were significantly (moderate effect) taller ($P = 0.013$, $d = 0.70$) and significantly (large effect) heavier ($P<0.001$, $d = 1.64$). A visual representation of the individual forward's and back's body mass values, along with the mean values and mean difference for each positional group, is presented via the Gardner–Altman plot in Fig 1.

All CMJ variables exhibited excellent relative between-trial reliability apart from countermovement phase time which demonstrated good-excellent reliability (Table 1). All outcome and propulsion phase variables showed excellent absolute between-trial reliability (Table 1). Absolute and relative countermovement displacement displayed good-excellent absolute between-trial reliability, whereas all remaining countermovement phase variables demonstrated good absolute between-trial reliability (Table 1). There were no significant differences in any countermovement phase or propulsion phase variable between forwards and backs with

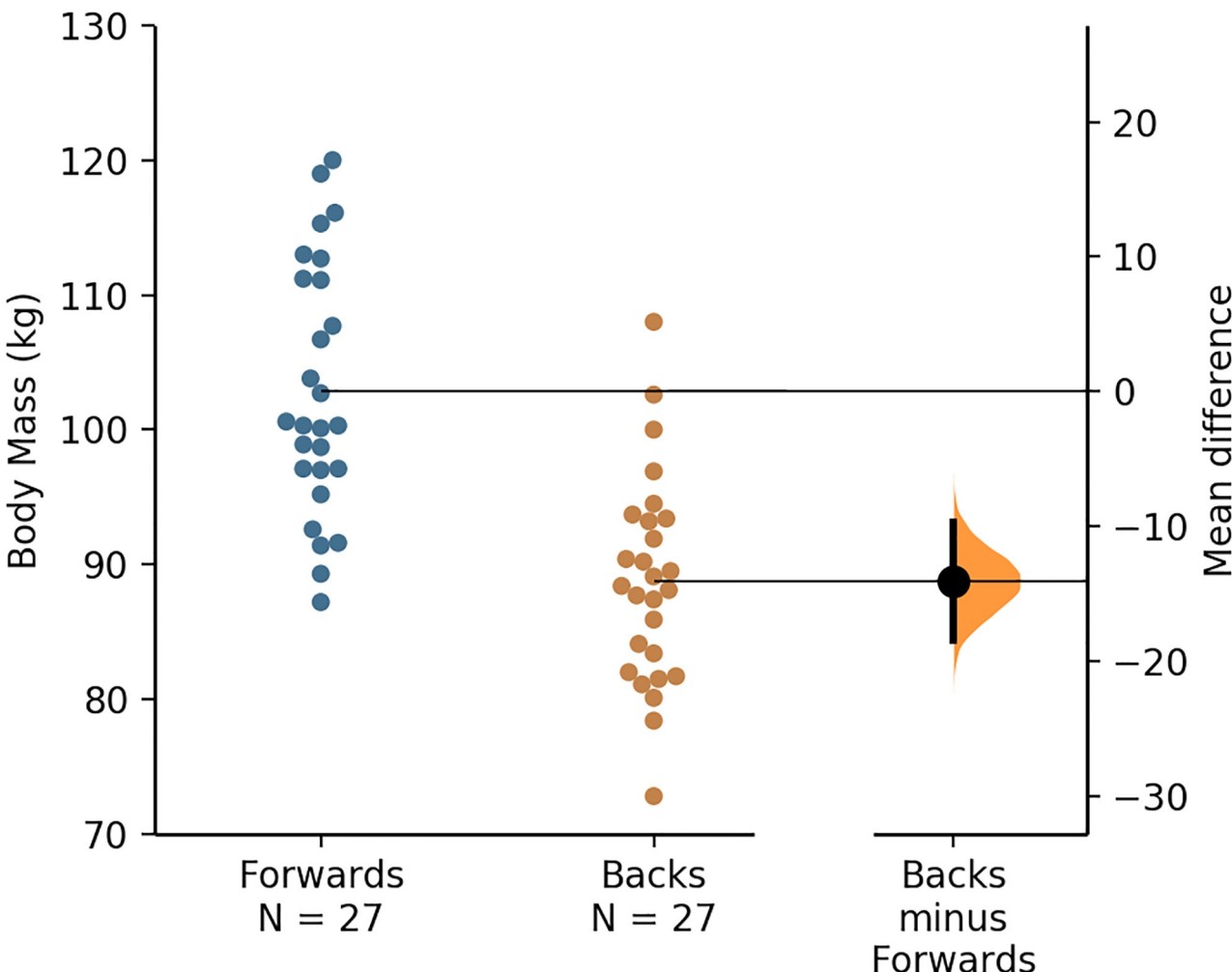

**Fig 1. A Gardner–Altman plot of the forwards' and backs' body mass data, with the mean and mean difference for and between each positional group, respectively.**

just small effect sizes noted (Table 1). Jump height was significantly greater for backs with moderate effects displayed (Table 1). Take-off momentum was largely and significantly greater for forwards (Table 1).

A visual representation of the individual forward's and back's jump height and take-off momentum values, along with the mean values and mean differences for each positional group, is presented via the Gardner–Altman plots in Figs 2 and 3.

The quadrant scatter chart that illustrates the cause-effect relationship between take-off momentum and body mass is presented in Fig 4. There was a very large relationship between take-off momentum and body mass ($r = 0.865$, $P < 0.001$), with body mass explaining ~75% of the variance take-off momentum values (Fig 4).

## Discussion

The primary aim of this study was to compare select force-time variables from the counter-movement and propulsion phases of the CMJ, in addition to typical (i.e., jump height) and alternative (i.e., take-off momentum) outcome variables, between the global forwards and

**Table 1. The mean (± standard deviation) countermovement jump outcome and strategy values attained by each position and associated reliability data.**

| Variable | Forwards | Backs | d | p | ICC | ICC LoCI₉₅ | ICC UpCI₉₅ | CV% | CV% LoCI₉₅ | CV% UpCI₉₅ |
|---|---|---|---|---|---|---|---|---|---|---|
| *Outcome* | | | | | | | | | | |
| Jump Height (m) | 0.34 ± 0.04 | 0.36 ± 0.04 | -0.60 | 0.03 | 0.96 | 0.94 | 0.98 | 3.23 | 2.39 | 4.06 |
| Reactive strength index modified (ratio) | 0.44 ± 0.08 | 0.48 ± 0.08 | -0.46 | 0.10 | 0.95 | 0.93 | 0.97 | 5.51 | 4.09 | 6.94 |
| Take-off Momentum (kg·m/s) | 265.11 ± 28.52 | 237.42 ± 26.24 | 1.01 | 0.00 | 0.99 | 0.99 | 1.00 | 1.58 | 1.19 | 1.98 |
| *Propulsion Phase* | | | | | | | | | | |
| Mean Net Force (N) | 997.67 ± 165.36 | 922.46 ± 158.89 | 0.46 | 0.09 | 0.97 | 0.96 | 0.98 | 3.72 | 2.68 | 4.77 |
| Rel. Mean Net Force (N/BW) | 0.99 ± 0.17 | 1.06 ± 0.17 | -0.40 | 0.51 | 0.97 | 0.95 | 0.98 | 3.72 | 2.68 | 4.77 |
| Time (s) | 0.266 ± 0.036 | 0.259 ± 0.037 | 0.20 | 0.47 | 0.96 | 0.94 | 0.98 | 3.39 | 2.36 | 4.43 |
| Displacement (m) | 0.429 ± 0.057 | 0.434 ± 0.059 | -0.09 | 0.76 | 0.96 | 0.94 | 0.98 | 3.48 | 2.51 | 4.44 |
| Rel. Displacement (%) | 23.53 ± 2.37 | 23.97 ± 3.19 | -0.15 | 0.57 | 0.95 | 0.92 | 0.97 | 3.48 | 2.54 | 4.41 |
| *Countermovement Phase* | | | | | | | | | | |
| Net Force at Zero Velocity (N) | 1429.19 ± 346.58 | 1336.67 ± 275.35 | 0.30 | 0.14 | 0.96 | 0.94 | 0.98 | 6.63 | 4.75 | 8.52 |
| Rel. Net Force at Zero Velocity (N/BW) | 1.43 ± 0.36 | 1.54 ± 0.31 | -0.33 | 0.51 | 0.97 | 0.95 | 0.98 | 6.63 | 4.75 | 8.52 |
| Time (s) | 0.615 ± 0.067 | 0.612 ± 0.090 | 0.04 | 0.87 | 0.90 | 0.84 | 0.94 | 7.21 | 5.18 | 9.23 |
| Displacement (m) | 0.318 ± 0.052 | 0.331 ± 0.061 | -0.22 | 0.44 | 0.96 | 0.94 | 0.98 | 4.32 | 2.95 | 5.68 |
| Rel. Displacement (%) | 17.49 ± 2.20 | 18.29 ± 3.35 | -0.28 | 0.76 | 0.95 | 0.93 | 0.97 | 4.32 | 3.08 | 5.55 |
| Peak Velocity (m/s) | 1.30 ± 0.23 | 1.31 ± 0.21 | -0.05 | 0.85 | 0.94 | 0.91 | 0.97 | 5.44 | 3.73 | 7.16 |

Negative effect sizes indicate that backs achieved a larger mean value than forwards. ICC = intraclass correlation coefficient. CV% = coefficient of variation percentage. LoCI₉₅ = lower 95% confidence interval. UpCI₉₅ = upper 95% confidence interval. Rel. = relative. BW = bodyweight.

backs positional groups within rugby league. Countermovement phase variables did not discriminate between forwards and backs (Table 1). Also, none of the propulsion phase variables were significantly different between positions (Table 1). Therefore, the hypothesis that absolute kinetic variables would be larger for the forwards and relative (to body mass) kinetic variables would be larger for the backs was rejected for all variables.

Differences in countermovement phase variables has distinguished between senior SL and senior RLC players [7] and between senior and academy RLC players [5] when combined data from forwards and backs were analysed collectively. The lack of significant between-position differences in countermovement phase variables in the present study could, therefore, be due to the inclusion of forwards and backs who competed at the same senior level (i.e., RLC) only. More research is needed to ascertain whether countermovement phase variables distinguish between forwards and backs who compete in rugby league at different levels. Additionally, countermovement phase variables obtained from the CMJ test tend to be more sensitive to acute neuromuscular fatigue [32] and so a worthwhile future research avenue may be to explore whether countermovement phase variables can be used to determine different neuromuscular fatigue responses for forward and backs. At the very least, select countermovement phase variables, such as those reported in the present study, should still be regularly monitored as part of routine CMJ testing of rugby league players as they help to explain maintenance or change in CMJ outcome variables.

The jump height (and thus take-off velocity) was significantly greater for backs (moderate effects), whereas the take-off momentum was largely and significantly greater for forwards (Table 1), which was in line with the hypothesis. The greater jump height values attained by the backs agrees with the results of a recent study [8]. The mean jump height values reported for the forwards and backs in the present study correspond to the 50th and 45th percentile, respectively, based on the norm-referenced values for high-level rugby league players reported

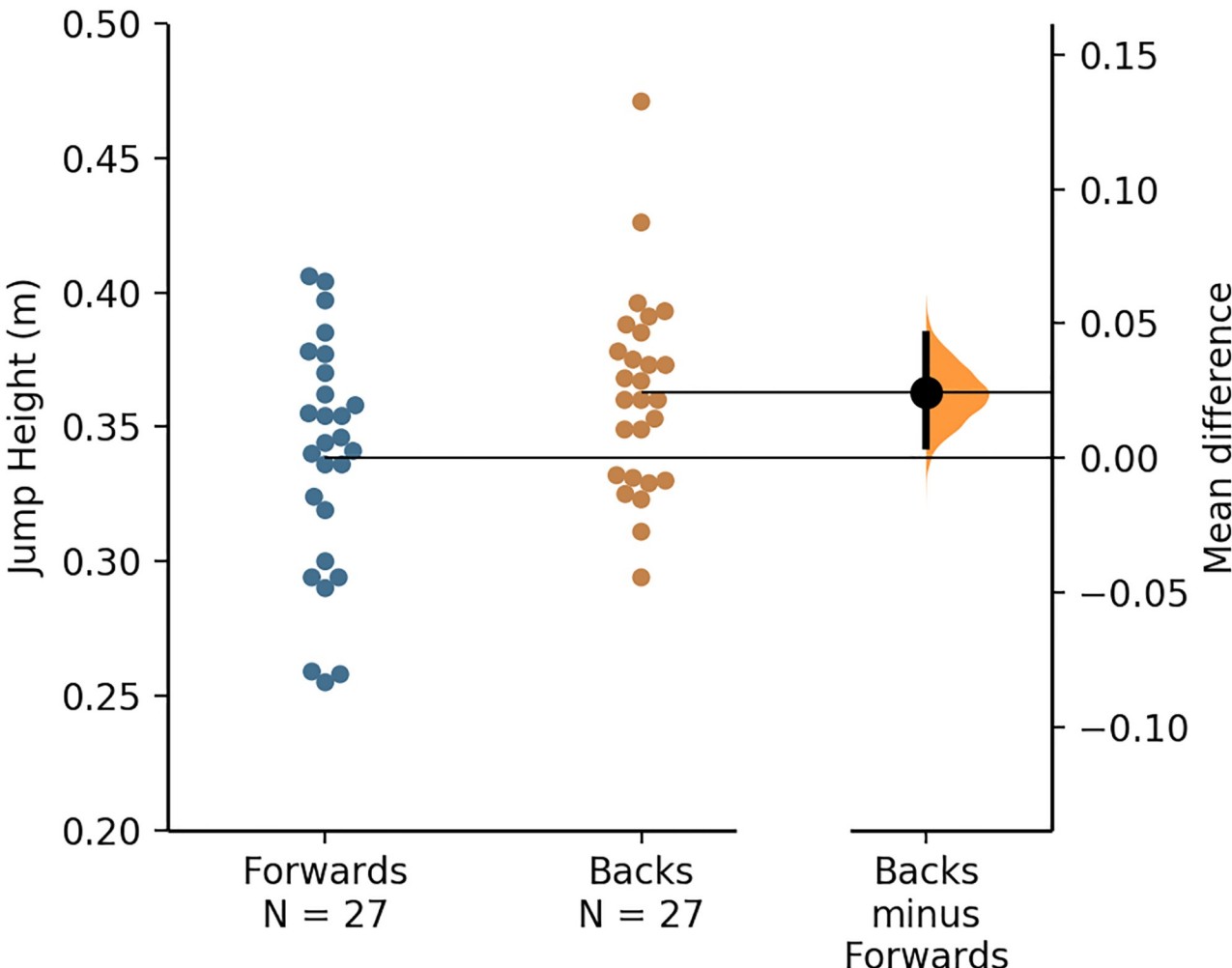

**Fig 2. A Gardner–Altman plot of the forwards' and backs' jump height data, with the mean and mean difference for and between each positional group, respectively.**

recently [8]. Owing to take-off velocity being higher for backs (as inferred from the higher jump height values), the larger take-off momentum (body mass × velocity) for the forwards (Table 1) was largely a consequence of their larger mass (Fig 1). The CMJ take-off momentum of rugby league players has only been reported in one study, but descriptive statistics were not provided [11]. However, in another study, researchers reported that the mean propulsion net impulse (which is identical to take-off momentum) attained by senior RLC players was 254 ± 28 Ns [5] which is very similar to the mean take-off momentum of 251 ± 31 kg·m/s attained by the RLC players who participated in the present study (Fig 4). Therefore, the jump height and take-off momentum values reported in this study may be considered a representative sample of high-level rugby league players' CMJ data.

In line with the above discussion point, there was a very large relationship between body mass and take-off momentum (Fig 4). As hypothesised, however, there was considerable overlap of individual player's body mass and CMJ variables such as jump height and take-off momentum across the forwards and backs (Figs 1–3), despite significant differences in the mean values attained by each positional group. Therefore, although the included participants

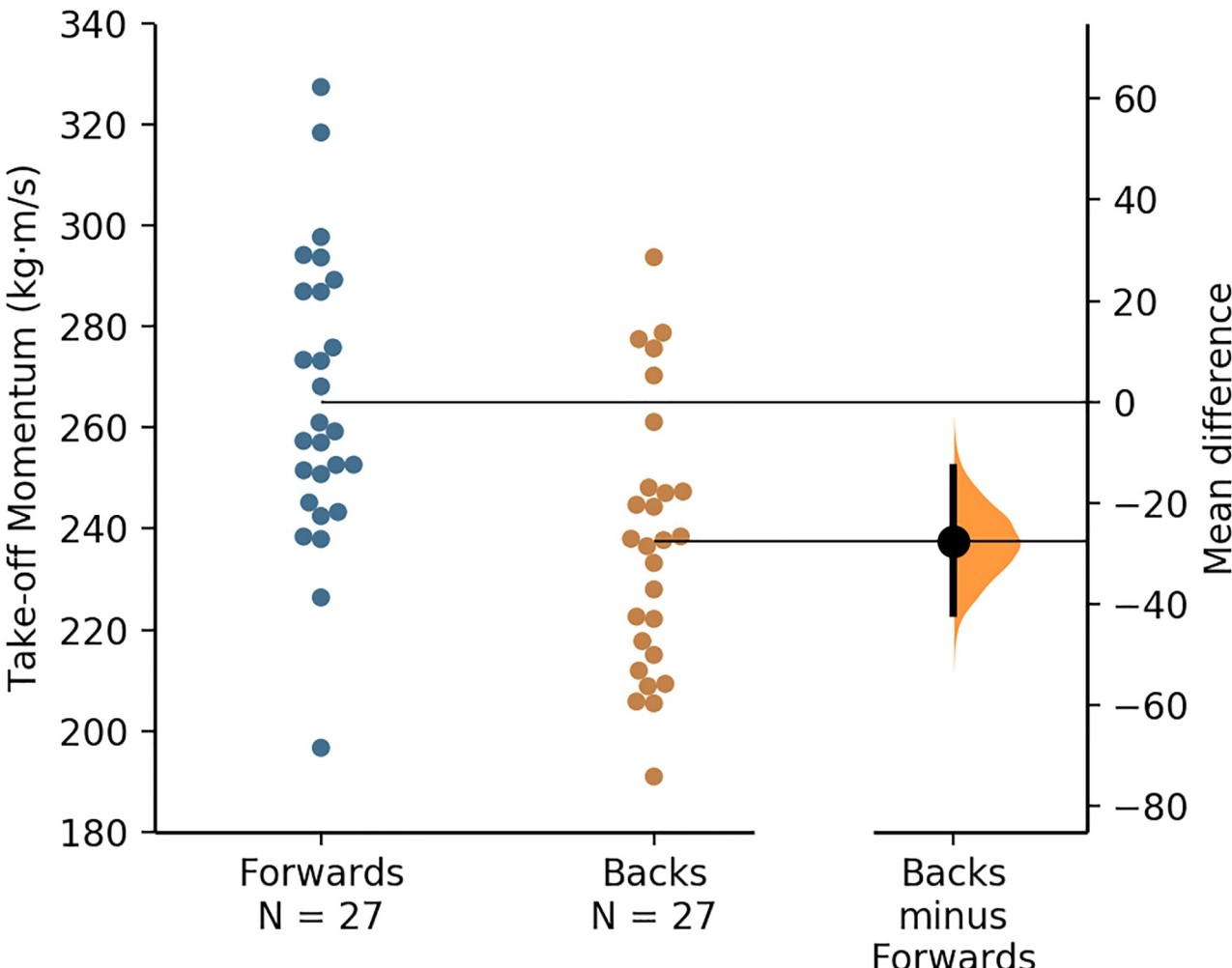

**Fig 3. A Gardner–Altman plot of the forwards' and backs' take-off momentum data, with the mean and mean difference for and between each positional group, respectively.**

were stratified into the global forwards and backs positional groups and significant positional differences in CMJ outcome variables were noted, ultimately rugby league is a collision sport that requires players from each positional group to be able to attain high movement velocities and utilise their body mass effectively during competition, for example, during tackles. However, the cause-effect relationship between body mass and take-off momentum, that is visually presented as a quadrant scatter plot in Fig 4, shows that a heavier body mass does not always contribute effectively to take-off momentum. For example, there was a cluster of forwards who had above average body mass when compared to the whole cohort, but their momentum was also below average (Fig 4). These players, therefore, fall into the undesirable high body mass with low momentum quadrant, located as the bottom-right quadrant in Fig 4.

Although only CMJ data are presented in this study, very large-near perfect associations between CMJ take-off momentum and sprint momentum were reported in the study that involved a high-level (RLC) rugby league cohort [11], probably attributable to the large variance in body mass and low variance in jump height for the involved cohort [12]. While correlation and causation are different, it is reasonable to suggest that the cluster of forwards with

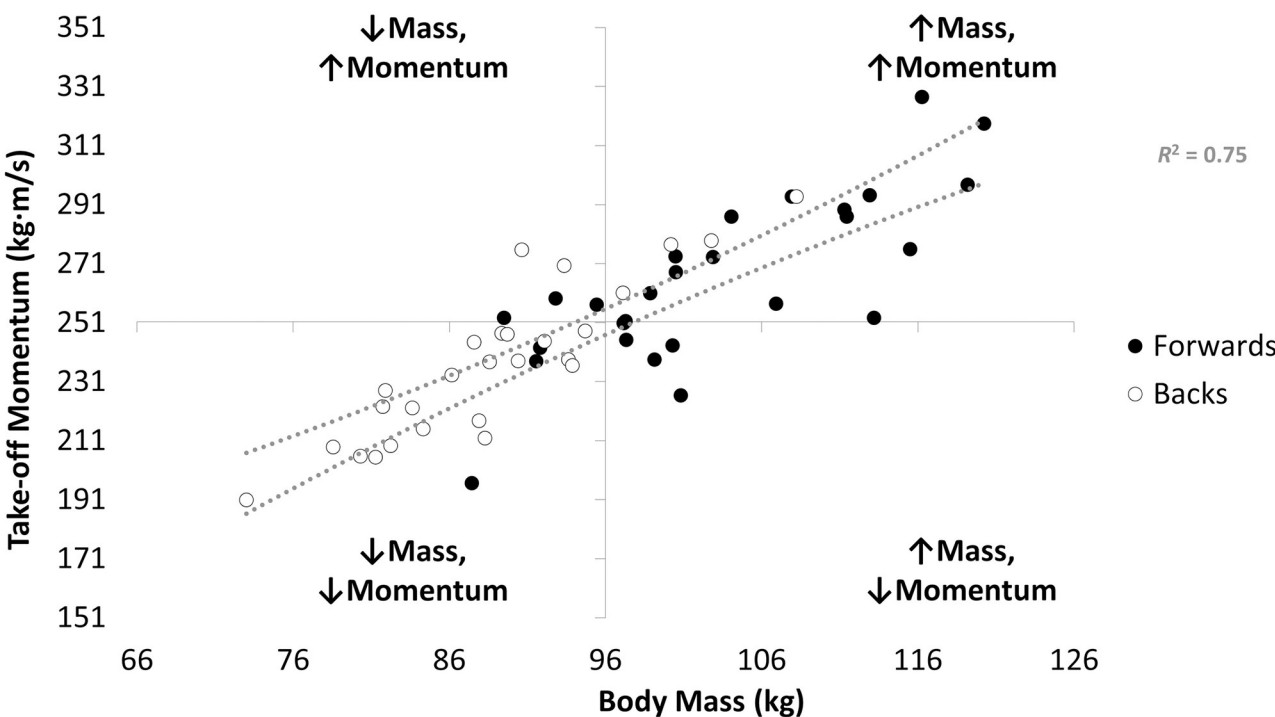

**Fig 4. A quadrant scatter chart illustrating the relationship between take-off momentum and body mass.** The forwards' data are represented by the black circles and the backs' data are represented by the open circles. The top and bottom dashed lines represent the upper and lower 95% confidence interval, respectively, of the relationship between take-off momentum and body mass.

high body mass, but low momentum (Fig 4), may also demonstrate lower than desirable sprint momentum. Sprint momentum has been highlighted as a key performance indicator in rugby league [10]. Forwards experience the most collisions during competition [33], particularly hit-up forwards [34, 35]. Therefore, the ability to attain a large momentum can be considered especially important for rugby league forwards which potentially places the forwards who fall into the high body mass with low momentum quadrant (Fig 4) at a performance disadvantage. As mentioned earlier though, players from each positional group would benefit from being able to move fast and make use of a large body mass during competition. Evidence of this includes the two backs who feature in the low body mass with high momentum quadrant. These players achieved a large take-off momentum in the CMJ by being able to attain a high take-off velocity and, therefore, a high jump.

## Conclusion

While the overarching aim of this study was to identify position-specific CMJ force-time variables within rugby league, the results suggest that it may be more beneficial to rugby league athletes and practitioners to identify player-specific, or at least sub-position-specific, variables. As a minimum, it would be worthwhile for the CMJ force-time variables to be selected based on what is considered important to individual player's or small clusters of similar players' successes during competition. For example, if a player's key area for competition success is driven by their capacity to effectively accelerate their own body mass, as they are typically involved in fewer collisions and cover a greater volume and distance of sprint running during a match, then CMJ height, relative mean net forces and phase times may be suitable variables of interest.

Alternatively, if a player's key area for competition success is dictated by their ability to effectively utilise their body mass to drive the opposition backwards, as they are typically involved in frequent collisions and cover a lower volume and distance of sprint running during a match, then CMJ take-off momentum and absolute mean net forces may be suitable variables of interest. If CMJ take-off momentum is to be reported (11), then plotting relationship between body mass and take-off momentum for the whole rugby league squad via a quadrant

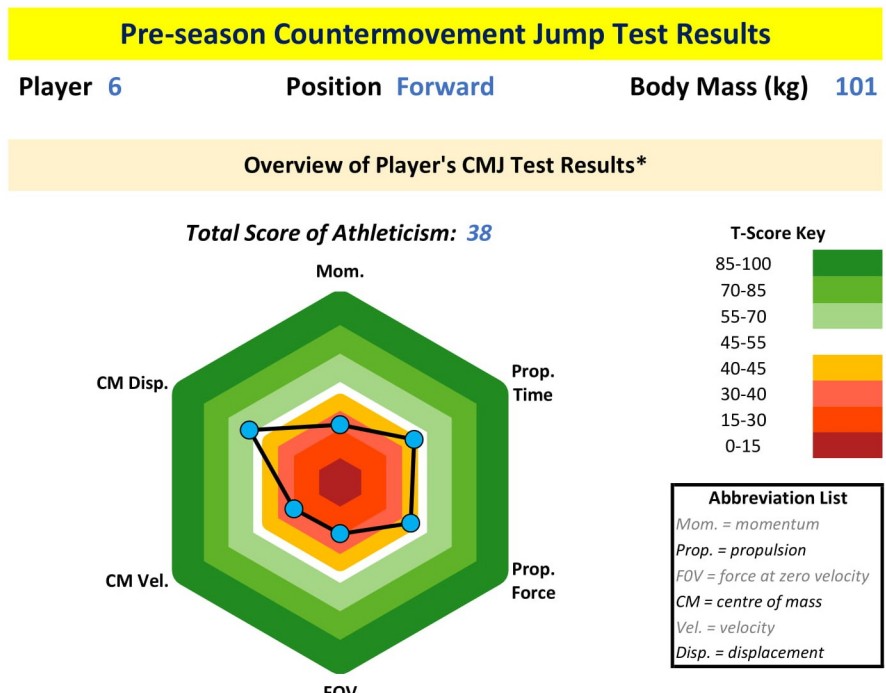

*Results have been converted to t-scores ranging from 0-100 (50 = forwards' average)

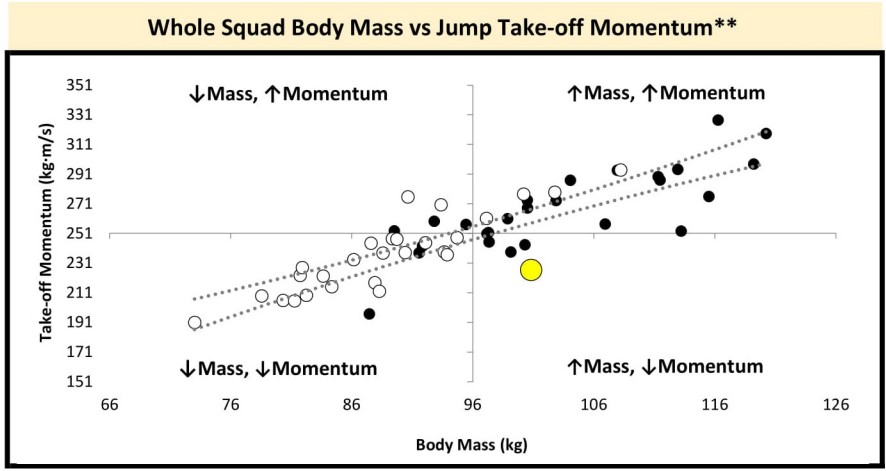

**Player's position on the the above graph is highlighted as the large yellow circle

**Fig 5. An example one-page report of countermovement jump variables attained by one of the rugby league forwards who participated in this study, alongside the body mass and take-off momentum relationship for the whole rugby league squad via a quadrant scatter chart (per Fig 4).** T-scores were calculated using squad median value due to sample size.

scatter chart should help to identify individual players who may benefit from either reducing body mass (in the form of fat-mass) or increasing their net force expression (i.e. increasing maximal strength), or a combination of both, to develop a larger momentum during jumping and sprinting. Two examples of how one may consider reporting key CMJ variables for specific players, alongside the body mass and take-off momentum relationship for the whole rugby league squad via a quadrant scatter chart, are presented as Figs 5 and 6.

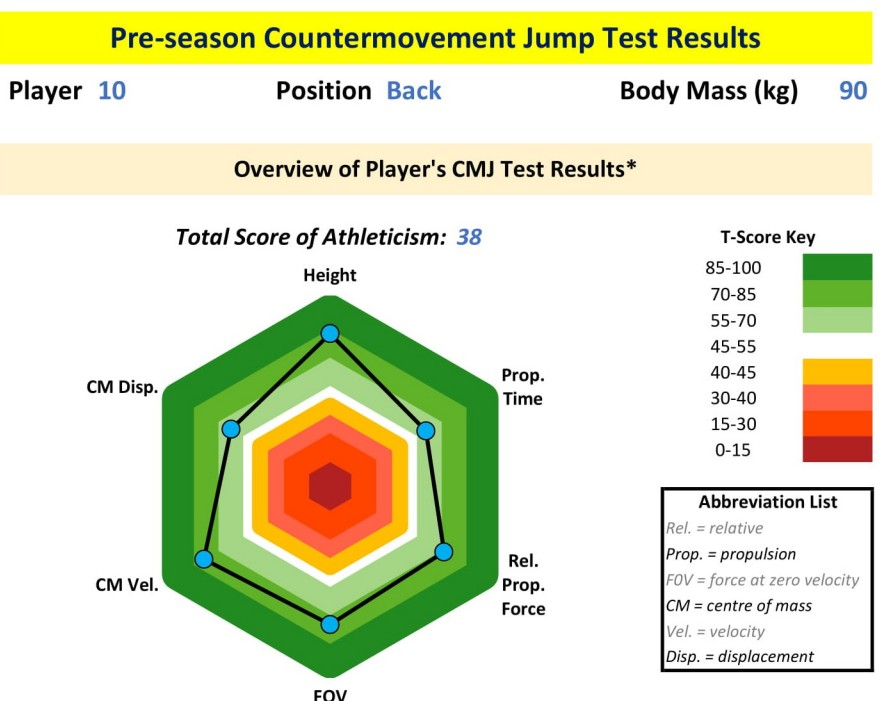

*Results have been converted to t-scores ranging from 0-100 (50 = backs' average)

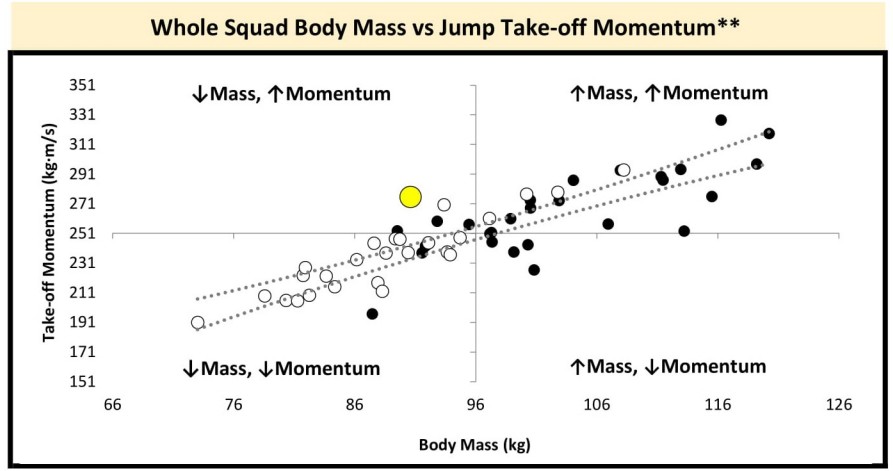

**Player's position on the the above graph is highlighted as the large yellow circle

**Fig 6. An example one-page report of countermovement jump variables attained by one of the rugby league backs who participated in this study, alongside the body mass and take-off momentum relationship for the whole rugby league squad via a quadrant scatter chart (per Fig 4).** T-scores were calculated using squad median value due to sample size.

## Supporting information

**S1 Table. The jump data for forwards (n = 27) and backs (n = 27) that was taken forward for statistical analyses.** RSImod = reactive strength index modified and BW = bodyweight. (XLSX)

## Acknowledgments

The authors would like to thank the participants of the study.

## Author Contributions

**Conceptualization:** John J. McMahon.

**Data curation:** John J. McMahon.

**Formal analysis:** John J. McMahon.

**Methodology:** John J. McMahon, Jason P. Lake, Paul Comfort.

**Visualization:** John J. McMahon.

**Writing – original draft:** John J. McMahon.

**Writing – review & editing:** John J. McMahon, Jason P. Lake, Paul Comfort.

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
