## [Decision Letter · Decision Letter 0]

4 Feb 2022

PONE-D-21-36555Identifying and reporting position-specific countermovement jump outcome and phase characteristics within rugby leaguePLOS ONE

Dear Dr. McMahon,

Thank you for submitting your manuscript to PLOS ONE. After careful consideration, we feel that it has merit but does not fully meet PLOS ONE’s publication criteria as it currently stands. Therefore, we invite you to submit a revised version of the manuscript that addresses the points raised during the review process.

We look forward to receiving your revised manuscript.

Kind regards,

Andrew Philip Lavender, PhD

Academic Editor

PLOS ONE

Journal Requirements:

Reviewers' comments:

Reviewer's Responses to Questions

**Comments to the Author**

1. Is the manuscript technically sound, and do the data support the conclusions?

Reviewer #1: Yes

Reviewer #2: Yes

2. Has the statistical analysis been performed appropriately and rigorously? 

Reviewer #1: Yes

Reviewer #2: Yes

3. Have the authors made all data underlying the findings in their manuscript fully available?

Reviewer #1: No

Reviewer #2: Yes

4. Is the manuscript presented in an intelligible fashion and written in standard English?

Reviewer #1: Yes

Reviewer #2: Yes

5. Review Comments to the Author

Reviewer #1: The aim of this study was to identify position-specific CMJ force-time variables within rugby league and, more specifically, to compare typical and alternative CMJ variables between the global forwards and backs positional groups. The work is perfectly written and is very practical for rugby league practitioners. I recommend some minor changes or suggestions before acceptance. I Congratulations to the authors, as usual, for their magnificent work.

Lines 39-42. It would be advisable to report the results with the corresponding statistical parameters for greater clarity.

Lines 78-79. Avoid repeating the words "however" in two consecutive sentences.

Lines 140-143. Justify the sample size used in this study.

Lines 151-152. Was any instruction given regarding the velocity of the countermovement?

Note that these types of instructions can affect some CMJ performance variables.

https://pubmed.ncbi.nlm.nih.gov/31359825/

Lines 166-170. I recommend that the authors look at two recent studies to justify jump start and takeoff times.

https://pubmed.ncbi.nlm.nih.gov/31711369/

https://www.tandfonline.com/doi/abs/10.1080/1091367X.2021.1872578

Lines 176-189. Why was the reactive strength index modified not calculated? This metric has been extensively discussed in the Introduction section. In fact, the reactive strength index modified has been recently recommended to evaluate the CMJ performance because this metric provides an understanding of both the outcome measure and jump strategy.

https://journals.lww.com/nsca-scj/Abstract/9000/A_Framework_to_Guide_Practitioners_for_Selecting.99138.aspx

Lines 198-204. Indicate which model was used to determine the ICCs and how the CV was calculated.

Reviewer #2: Thank you for allowing me to review the following manuscript. I really enjoyed reading it and feel all of my comments/ suggestions can be easily addressed.

Abstract

Line 28 – When discussing “position-specific” throughout, can we get the word “field” or something similar to avoid confusion. I understand that you mean position on the field, however, they way it is written it could be interpreted as position on the force plate or in the CMJ.

Introduction

Line 63-65: I like the use of the effect size descriptors; however, can you consider re-wording the below sentence. Particularly the part “…just a trivially greater CMJ height,”

“Conversely, senior players who competed in the English Super League (SL) achieved just a trivially greater CMJ height (d=0.12), but a moderately greater mean propulsion power (d=0.94) than the u19s players’ scores (6).”

Line 66: Change “these data were” to “this data was” as you are only talking about one aspect of data (propulsive phase time) ???

Line 67-69: Apologies for my lack of knowledge in terms of the English RL competition, but I’m assuming SL is a higher level of competition than the RLC? Can this be mentioned somewhere so it is clear you are comparing two different playing levels?

Line 70 – What do you mean by “Rugby league global positions standpoint...”

Paragraph beginning at line 70 – Just wanted to make comment that this is a fantastic paragraph that really highlights the need to look at positions individually. Really enjoyed it!

Line 93-94: Why is the correlation score presented as a range? (r = 0.78- 92; 61-85% shared variance [R 2 94 ]). Is it each individual’s correlation between jump to momentum and sprint momentum? Could this be explained or potentially just report the total mean r score and associated correlation descriptor.

Line 131 – “…owing to the larger body mass of the former.”

I’m not sure which is the former as they are both mentioned in this sentence, can this sentence be re-worded to be clear?

Methods & Materials

Line 190-193: Why did you look at both Vertical velocity at T/O and jump height as variables? Although they provide different numbers, are they not essentially the same thing?? (As you can equate JH from VEL @ take-off).

Statistical analyses

Line 216 – Close bracket after small correlation descriptor e.g. )

Results

Figure 4 – Include a key for forwards and backs as opposed to explaining in the text. Just an idea

Discussion

Line 279-280: With a p-value of 0.09, I don’t think this is significant. Please change this sentence.

“Propulsion mean net force was significantly greater for forwards, albeit with only a small effect shown (Table 1)…”

Line 287 – Change significant to “significance”

Line 297 – Delete routine or change to avoid repetitive routinely/ routine

Line 299- I reiterate the point that I don’t see the need to report both Take-off velocity and Jump height (they are the same thing). I would suggest picking one and removing any text/ discussion of the other. I would suggest using jump height, as it is probably easier to compare to other studies using CMJs.

Note: I really like the inclusion of figure 4 and feel this is very novel. Great work!

Conclusion:

For the sake of a sentence, I’m not sure if figure 5 and 6 are required. I appreciate that it links the research to practice (which is important), but I feel what is in the text is a good enough suggestion on its own. If you need to cut out any particular part of the manuscript, I feel these don’t add a lot.

6. PLOS authors have the option to publish the peer review history of their article (what does this mean?). If published, this will include your full peer review and any attached files.

Reviewer #1: **Yes: **Alejandro Pérez-Castilla

Reviewer #2: **Yes: **Mathew William O'Grady

---

## [Author Response · Author response to Decision Letter 0]

7 Mar 2022

Reviewer #1: The aim of this study was to identify position-specific CMJ force-time variables within rugby league and, more specifically, to compare typical and alternative CMJ variables between the global forwards and backs positional groups. The work is perfectly written and is very practical for rugby league practitioners. I recommend some minor changes or suggestions before acceptance. I Congratulations to the authors, as usual, for their magnificent work.

Response: Thank you very much for your kind and encouraging comments.

Lines 39-42. It would be advisable to report the results with the corresponding statistical parameters for greater clarity.

Response: Thank you for your comment. We have now included the P and d values where required in the abstract.

Lines 78-79. Avoid repeating the words "however" in two consecutive sentences.

Response: Thank you for your comment. We have change the second instance to “nevertheless”.

Lines 140-143. Justify the sample size used in this study.

Response: Thank you for your comment. An a-priori statistical power calculation was not performed prior to conducting the study (to inform the sample size) owing to the lack of existing comparable data to inform this procedure. As mentioned in the introduction section, no study had compared the strategy and outcome variables for the CMJ between rugby league forwards and backs until now. So, there was no direct comparison from the scientific literature to draw upon to estimate likely effect sizes between positions. It is recommended to not perform post-hoc statistical power calculations despite this being common practice in sports-related studies. Therefore, we have no evidence to support the sample size included in the study. We added to the start of the methods section that a convenience sampling approach was adopted in the study to make it clearer how the sample was recruited. Beyond this, we are unsure what else, if anything, the reviewer would like us to add to the study regarding the included sample. 

Lines 151-152. Was any instruction given regarding the velocity of the countermovement?

Note that these types of instructions can affect some CMJ performance variables.

https://pubmed.ncbi.nlm.nih.gov/31359825/

Response: Thank you for your comment. The following statements have been added to the methods section – “The participants were informed that the “jump as fast” part of the verbal cue referred to them performing the countermovement and propulsion phases of the jump as quickly as possible. Verbal cues were standardised owing to their influence of CMJ force-time characteristics (16-17).” Please note that one of the two studies cited in this new sentence is the one you suggested and the other one is also relevant. These have been highlighted in the reference list too. 

Lines 166-170. I recommend that the authors look at two recent studies to justify jump start and takeoff times.

https://pubmed.ncbi.nlm.nih.gov/31711369/

https://www.tandfonline.com/doi/abs/10.1080/1091367X.2021.1872578

Response: Thank you for this suggestion. The jump start study you shared the link to involved externally loaded CMJs which would likely influence the 5SD approach used in our study (greater signal noise during quiet standing when loaded) so instead we have clarified in our study that we followed the protocol for unloaded CMJs (as highlighted where mentioned in the methods section). We did, however, add the take-off threshold study to the citations provided next to where the method is explained (specifically, it is study 23). 

Lines 176-189. Why was the reactive strength index modified not calculated? This metric has been extensively discussed in the Introduction section. In fact, the reactive strength index modified has been recently recommended to evaluate the CMJ performance because this metric provides an understanding of both the outcome measure and jump strategy.

https://journals.lww.com/nsca-scj/Abstract/9000/A_Framework_to_Guide_Practitioners_for_Selecting.99138.aspx

Response: Thanks for your comment. We originally did not include it because we present the jump height and the countermovement and propulsion phase times which is effectively unpacking the reactive strength index modified into its constituent parts. However, you raise a good point in that we discuss it a lot in the introduction, so we have added it to the results and remove take-off velocity based on reviewer 2’s suggestion. 

Lines 198-204. Indicate which model was used to determine the ICCs and how the CV was calculated.

Response: Thank you for your comment. We have added the ICC type in parentheses like so “[type 3,k]”.

Reviewer #2: Thank you for allowing me to review the following manuscript. I really enjoyed reading it and feel all of my comments/ suggestions can be easily addressed.

Response: Thank you very much for your kind and encouraging comments.

Abstract

Line 28 – When discussing “position-specific” throughout, can we get the word “field” or something similar to avoid confusion. I understand that you mean position on the field, however, they way it is written it could be interpreted as position on the force plate or in the CMJ.

Response: Thank you for your comment. We have amended the wording as follows “…identifying force platform-derived CMJ variables that may be more applicable to RL positions (e.g., forwards and backs)…”

Introduction

Line 63-65: I like the use of the effect size descriptors; however, can you consider re-wording the below sentence. Particularly the part “…just a trivially greater CMJ height,”

“Conversely, senior players who competed in the English Super League (SL) achieved just a trivially greater CMJ height (d=0.12), but a moderately greater mean propulsion power (d=0.94) than the u19s players’ scores (6).”

Response: Thank you for your comment. We have amended the wording as follows “…achieved a slightly greater CMJ height (d=0.12 [trivial effect])…”.

Line 66: Change “these data were” to “this data was” as you are only talking about one aspect of data (propulsive phase time) ???

Response: Thank you for your comment. We have amended this in line with your suggestion. 

Line 67-69: Apologies for my lack of knowledge in terms of the English RL competition, but I’m assuming SL is a higher level of competition than the RLC? Can this be mentioned somewhere so it is clear you are comparing two different playing levels?

Response: Thank you for your comment. We have clarified the level of competition when we first introduce each tier, for example “…(RLC, second tier of English Rugby League)…”.

Line 70 – What do you mean by “Rugby league global positions standpoint...”

Response: Thank you for your comment. We have amended this to “With respect to the two main positional groups within rugby league..”.

Paragraph beginning at line 70 – Just wanted to make comment that this is a fantastic paragraph that really highlights the need to look at positions individually. Really enjoyed it!

Response: Thank you very much for your kind and encouraging comment.

Line 93-94: Why is the correlation score presented as a range? (r = 0.78- 92; 61-85% shared variance [R 2 94 ]). Is it each individual’s correlation between jump to momentum and sprint momentum? Could this be explained or potentially just report the total mean r score and associated correlation descriptor.

Response: Thank you for your comment. We have added more detail to this sentence which hopefully now means that it makes sense (specifically we added, “..attained at 5, 10 and 20 m..”).

Line 131 – “…owing to the larger body mass of the former.”

I’m not sure which is the former as they are both mentioned in this sentence, can this sentence be re-worded to be clear?

Response: Thank you for your comment. We have changed “former” to “forwards” to make this point clearer.

Methods & Materials

Line 190-193: Why did you look at both Vertical velocity at T/O and jump height as variables? Although they provide different numbers, are they not essentially the same thing?? (As you can equate JH from VEL @ take-off).

Response: This is a fair point, and you are right that it tells the same story as jump height. We included it because it, alongside body mass, shows how the jump take-off momentum was achieved, but we agree the jump height does the job well enough on its own. So, we have removed it and replaced it in the Table with reactive strength index modified following reviewer 1’s suggestion. 

Statistical analyses

Line 216 – Close bracket after small correlation descriptor e.g. )

Response: That’s a great spot, thank you. We’ve corrected this. 

Results

Figure 4 – Include a key for forwards and backs as opposed to explaining in the text. Just an idea

Response: Thank you for this suggestion. We have added a key to the figure. 

Discussion

Line 279-280: With a p-value of 0.09, I don’t think this is significant. Please change this sentence.

“Propulsion mean net force was significantly greater for forwards, albeit with only a small effect shown (Table 1)…”

Response: Thank you for your comment. I’m not sure why we misinterpret this difference as being significant when it isn’t. We have since amended the results and discussion sections accordingly, for example:

“Also, none of the propulsion phase variables were significantly different between positions (Table 1). Therefore, the hypothesis that absolute kinetic variables would be larger for the forwards and relative (to body mass) kinetic variables would be larger for the backs was rejected for all variables.”

Line 287 – Change significant to “significance”

Response: Thank you for this suggestion. We have amended to “significance in positional…” to accommodate your suggestion but ensure that the intended meaning remained as we originally intended. 

Line 297 – Delete routine or change to avoid repetitive routinely/ routine

Response: Thank you for this suggestion. We have changed “routinely” to “regularly” 

Line 299- I reiterate the point that I don’t see the need to report both Take-off velocity and Jump height (they are the same thing). I would suggest picking one and removing any text/ discussion of the other. I would suggest using jump height, as it is probably easier to compare to other studies using CMJs.

Response: Per our earlier response to your comment about this, we have removed the take-off velocity data from the study. 

Note: I really like the inclusion of figure 4 and feel this is very novel. Great work!

Response: Thank you very much for your kind and encouraging comment.

Conclusion:

For the sake of a sentence, I’m not sure if figure 5 and 6 are required. I appreciate that it links the research to practice (which is important), but I feel what is in the text is a good enough suggestion on its own. If you need to cut out any particular part of the manuscript, I feel these don’t add a lot.

Response: Thank you for your comment. While we understand your perspective, we would like to keep these figures in the manuscript to show readers examples of how they may report this type data to athletes, as there are not many resources available to guide practitioners on how to report such results. We hope you understand.

---

## [Editor Report · Decision Letter 1]

14 Mar 2022

Identifying and reporting position-specific countermovement jump outcome and phase characteristics within rugby league

PONE-D-21-36555R1

Dear Dr. McMahon,

We’re pleased to inform you that your manuscript has been judged scientifically suitable for publication and will be formally accepted for publication once it meets all outstanding technical requirements.

Kind regards,

Andrew Philip Lavender, PhD

Academic Editor

PLOS ONE
---

## [Editor Report · Acceptance letter]

18 Mar 2022

PONE-D-21-36555R1 

Identifying and reporting position-specific countermovement jump outcome and phase characteristics within rugby league 

Dear Dr. McMahon:

I'm pleased to inform you that your manuscript has been deemed suitable for publication in PLOS ONE. Congratulations! Your manuscript is now with our production department. 

Kind regards, 

on behalf of

Dr. Andrew Philip Lavender 

Academic Editor

PLOS ONE